# Progressive Metabolic Dysfunction and Nutritional Variability Precedes Necrotizing Enterocolitis

**DOI:** 10.3390/nu12051275

**Published:** 2020-04-30

**Authors:** Tiffany J. Sinclair, Chengyin Ye, Yunliang Chen, Dongyan Zhang, Tian Li, Xuefeng Bruce Ling, Harvey J. Cohen, Gary M. Shaw, David K. Stevenson, Donald Chace, Reese H. Clark, Karl G. Sylvester

**Affiliations:** 1Department of Surgery, Stanford University School of Medicine, Stanford, CA 94305, USA; tjsin@stanford.edu (T.J.S.); Cyl_king@hotmail.com (Y.C.); dongyan2@stanford.edu (D.Z.); ddsnack@gmail.com (T.L.); bxling@stanford.edu (X.B.L.); 2Division of Pediatric Surgery, Stanford University School of Medicine, Stanford, CA 94305, USA; 3School of Medicine, Hangzhou Normal University, Hangzhou 311121, China; yechengyin@hznu.edu.cn; 4School of Computer Science, China University of Geosciences, Wuhan 430074, China; 5School of Computer and Communication Engineering, University of Science and Technology Beijing, Beijing 100083, China; 6Department of Pediatrics, Stanford University School of Medicine, Stanford, CA 94305, USA; harvey.cohen@stanford.edu (H.J.C.); gmshaw@stanford.edu (G.M.S.); david.stevenson@stanford.edu (D.K.S.); 7Medolac Laboratories, Boulder City, NV 89005, USA; dchace@medolac.com; 8Pediatrix-Obstetrix Center for Research, Education and Quality, Sunrise, FL 33323, USA; reese_clark@mednax.com; 9Stanford Metabolic Health Center, Stanford Children’s Hospital, Stanford, CA 94304, USA

**Keywords:** necrotizing enterocolitis, growth velocity, growth faltering, newborn calories, newborn metabolic profiling, metabolomics, prematurity, very low birthweight

## Abstract

Necrotizing Enterocolitis (NEC) is associated with prematurity, enteral feedings, and enteral dysbiosis. Accordingly, we hypothesized that along with nutritional variability, metabolic dysfunction would be associated with NEC onset. **Methods:** We queried a multicenter longitudinal database that included 995 preterm infants (<32 weeks gestation) and included 73 cases of NEC. Dried blood spot samples were obtained on day of life 1, 7, 28, and 42. Metabolite data from each time point included 72 amino acid (AA) and acylcarnitine (AC) measures. Nutrition data were averaged at each of the same time points. Odds ratios and 95% confidence intervals were calculated using samples obtained prior to NEC diagnosis and adjusted for potential confounding variables. Nutritional and metabolic data were plotted longitudinally to determine relationship to NEC onset. **Results:** Day 1 analyte levels of alanine, phenylalanine, free carnitine, C16, arginine, C14:1/C16, and citrulline/phenylalanine were associated with the subsequent development of NEC. Over time, differences in individual analyte levels associated with NEC onset shifted from predominantly AAs at birth to predominantly ACs by day 42. Subjects who developed NEC received significantly lower weight-adjusted total calories (*p* < 0.001) overall, a trend that emerged by day of life 7 (*p* = 0.020), and persisted until day of life 28 (*p* < 0.001) and 42 (*p* < 0.001). **Conclusion:** Premature infants demonstrate metabolic differences at birth. Metabolite abnormalities progress in parallel to significant differences in nutritional delivery signifying metabolic dysfunction in premature newborns prior to NEC onset. These observations provide new insights to potential contributing pathophysiology of NEC and opportunity for clinical care-based prevention.

## 1. Introduction

Necrotizing enterocolitis (NEC) is an acquired disease of premature neonates with a multifactorial pathophysiology that is likely influenced by both intrinsic biological factors (i.e., prematurity) and extrinsic variables including exposures inherent to medical care (i.e., nutrition).

Recent evidence has demonstrated that fatty acid and protein metabolism differs significantly between premature and term infants [1,2,3]. Metabolic differences associated with prematurity are further affected by feeding practices [3,4] and illness [2,5,6]. Standardized feeding guidelines have been associated with a lower risk of NEC in very low birth weight infants [7] and early nutrition has been shown to improve outcomes and decrease mortality [8]. The etiologic intersection between prematurity, nutrition, and NEC plausibly lies at the level of cellular metabolism. Prematurity may be associated with mitochondrial dysfunction as reflected in newborn screening and may confer a form of development-associated metabolic vulnerability [9,10]. In combination with extrinsic stressors (i.e., altered microbiome, direct toxicity of nutritional macromolecules) metabolic vulnerability could lead to abnormal nutrient metabolism, disruptions in cellular energy balance, and oxidative stress initiating the pathophysiologic cascade that results in NEC [11,12,13].

Newborn screening (NBS), which includes measurements of amino acids (AAs) and acylcarnitines (ACs)—intermediates of protein and fatty acid metabolism respectively—was initially implemented for the early identification of inborn errors of metabolism. Our group has previously demonstrated an association between levels of specific ACs measured on NBS at birth and the subsequent development of NEC in preterm infants. [14] We therefore sought to determine if serial NBS panels could be utilized to elucidate the relationship between prematurity associated metabolic dysfunction and nutrition on the development of NEC. We hypothesized that progression in measured changes of intermediate metabolites (AA and AC) together with clinical nutrition could be utilized to determine if metabolic dysfunction precedes the onset of NEC.

## 2. Materials and Methods

We performed a case-control study derived from querying a longitudinal multicenter (23 sites in 17 states) database collected by the Pediatrix-Obstetrix Center for Research, Education and Quality across their network of neonatal intensive care units (NICUs) from April, 2009 to September, 2012 [15]. These data were comprised of 995 preterm infants (<32 weeks gestation) with four serial metabolite samples collected over the first six weeks of life in the NICU. Infants with chromosomal or congenital anomalies were excluded.

Data on intestinal disease were captured in a structured field (choices were: none, isolated intestinal perforation, NEC-medical, NEC-surgical, and unknown). NEC was determined to be present if an infant had at least one of the following clinical signs: bilious gastric aspirate or emesis, abdominal distention, or blood in stool without evidence of a rectal fissure; and had one or more of the following radiographic findings: pneumatosis intestinalis, hepatobiliary gas, or pneumoperitoneum. Infants with a diagnosis of NEC-medical who subsequently were treated with surgery (had both diagnoses) were assigned to the NEC-surgical group. Of 76 cases of NEC, three exited the study early due to being transferred and were excluded from analysis. A total of 73 infants with NEC were included as cases. Cases of NEC were further sub-grouped based on day of diagnosis: post-delivery day 1–7 (*n* = 8), 8–28 (*n* = 48), 29–42 (*n* = 9), and >42 (*n* = 8). Control infants were selected from the database based on the following criteria: (1) alive at the end of the study and (2) no history of gastrointestinal disease (*n* = 814).

Detailed demographic and clinical data were collected for each case and control infant. Standardized fluid management and nutritional guidelines were utilized at each site. In brief, the total fluid infusion rate was set between 90 and 120 mL/kg/day. Glucose infusions were set to start immediately in critically ill premature neonates and infused at a rate of up to 8 mg/kg/min. Amino acid supplementation was started between 0.5 and 3.0 g/kg/day as early as possible and advanced by 0.5–1 g/kg/day to a maximum of 3.5 g/kg/day. Amino acid supplementation decreased to 1–2 g/kg/day once feedings reached 80–100 mL/kg/day. There were three amino acid formulations in use across the collection sites including, Aminosyn^®^ (Hospira), Premasol^®^ (Baxter) and TrophAmine^®^ (Braun). The utilization of these varied insignificantly among the case and control cohorts. Qualitatively, those amino acids that were identified as significantly different between the cases and controls (Table 1, D1 and D7), are not significantly different in each of the utilized AA formulations—Premasol^®^ and TrophAmine^®^ have identical concentrations (minor differences in Aminosyn^®^) for each of the essential and non-essential AAs reported as statistically different post adjustment in Table 1 and Figure 1 and Figure 2.

Lipids emulsions were initiated with parental nutrition at a rate of 1 g/kg/day and advanced by 0.5 g/kg/day to a maximum of 3.5 g/kg/day. Qualitatively, soy based Intralipid^®^ 10%(Baxter) was utilized throughout and quantitative differences in utilization did not emerge until D28 and reflects an increase use of IL in the cases to balance the deficiency in enteral calories achieved. Low rate, small volume (<10 mL/kg/day) enteral feedings were begun within the first week of life. Feedings could be advanced at a rate between 10 and 30 mL/kg/day. Full feedings were defined as 130 mL/kg/day with consistent weight gain of 20 to 30 g/kg/day. Human breast milk was the preferred form of nutrition. Supplements such as human milk fortifier (HMF) and multi chain triglycerides (MCT) could be added to feedings as per site preference by the time that feedings had reached 100 mL/kg/day. Nutritional data were averaged over the following time intervals: day of birth (day 1), 2–7, 8–28, and 29–42. All infants were followed until discharge from hospital.

Patient samples were collected within 24 h after birth (day 1), on approximately day 7 (7–8), on day 28 (27–29), and on day 42 (41–43) post-delivery or at discharge, whichever came first [15]. Dried blood spots from each time point were assayed for a panel of metabolites representing standard newborn screening and analysis included 14 AAs, 9 AA ratios, 35 ACs, 12 AC ratios, and two combined AA + AC measures. All testing was performed at the time of collection at a central laboratory (PerkinElmer Genetics, Bridgeville, PA, USA) and samples were analyzed using tandem mass spectrometry.

Patient characteristics were compared between case infants with NEC and control infants using *t*-test for quantitative variables and Fisher’s exact test for categorical variables.

Odds ratios (OR) and 95% confidence intervals (CIs) were calculated for all NEC cases using day 1 analyte levels, as well as at each time point of sampling for each sub-group. For the day 7, day 21, and day 42 time point analyses, only samples obtained prior to NEC diagnosis were used for the subjects in the NEC group. Thus, the analysis at each time-point represents only those differences between cases and controls that emerged prior to a diagnosis of NEC. This same approach was used for subsequent pre-NEC longitudinal analyses. Patient characteristic variables that were found to be significantly different in univariate analysis (Table 2) were considered as potential confounding variables and adjusted for in models. The clinical variables that were adjusted for included the following: birth weight and gestational age, transfused (Y/N), postnatal steroids (Y/N), late onset sepsis, and retinopathy of prematurity. Both unadjusted and adjusted logistic regression models were run individually for all analytes.

To identify those metabolic changes that were associated with the development of NEC over time, we first derived each metabolite’s fold-changes between cases and controls at each time point respectively (i.e., day 1, day 7, day 28, day 42). Second, we obtained the test statistic χ^2^ (Equation (1)) for each metabolite across the four time points in terms of the fold change of the *i*th metabolite at the *j*th time point *x_i_*_,*j*_, average fold change at the *j*th time point μ*_j_*, and the standard deviation of each time point σ*_j_* (Equation (2)):χ_*i*_^2^ = Σ_*j*_*Z*_*i*,*j*_^2^(1)
*Z*_*i*,*j*_ = (*x*_*i*,*j*_ − μ_*j*_)/σ_*j*_(2)

The significant metabolites were defined as those having p-values of χ^2^ (*df* = 4) < 5.62 × 10^−4^ after Bonferroni correction.

Loess curves were plotted over time for the nutrition data. Each nutrition variable was compared between NEC cases and their controls for each time point (day 1, 7, 28, 42) using one-tailed *t*-tests in a post-hoc analysis. The individual time point analysis was also conducted using only information obtained prior to NEC diagnosis, as detailed above. The proportion of enteral feedings given as human breast milk was compared at each time point using a one-tailed *t*-test. Weight gain velocity was calculated for each of the time points using Equation (3):Weight gain velocity (g/kg/day) = 1000 × ((weight*_t_*_+1_ − weight*_t_*)/weight_t_)/((*t* + 1) − *t*)(3)
where *t* represents the day of delivery at a beginning of a study time interval (i.e., day of delivery 1) and *t* + 1 represents the day of delivery at the end of a study time interval (e.g., day of delivery 7) [16,17].

All statistical analyses were performed using *R*. Results were reported as significant if the *p*-value was <0.05.

## 3. Results

Patient characteristics are summarized in Table 2. There was no significant difference in sex or race/ethnicity between NEC case and control infants. Infants who developed NEC were significantly more premature and had significantly lower birth weights. Infants who developed NEC were more likely to have received a blood transfusion and post-natal IV steroids, and more likely to be have been diagnosed with blood-culture positive sepsis or retinopathy of prematurity (ROP) (any stage). The mean age of NEC diagnosis was 20.5 days of life. Of the 73 infants with NEC, 48 (66%) were diagnosed with medical NEC and 25 (34%) were diagnosed with surgical NEC. Fourteen infants with NEC (19%) died at a mean post-natal age of 29.8 days (IQR 15.5, 30.0).

After adjusting for potential confounders, those metabolites that were found to have a statistically significant association with developing NEC are summarized in Table 1. On days 1 and 7, metabolic differences between infants with NEC and controls involved predominantly AAs, whereas on days 28 and 42, differences were exclusively noted for ACs. Despite the use of three different amino acids formulations, we were not able to detect an independent impact on the reported findings.

The longitudinal relationship between case and control metabolic changes are summarized as a heat map with hierarchical clustering in Figure 1. At each time point, the ratio of the fold changes is depicted. Several individual metabolites or metabolite ratios were found to have significant associations with NEC upon longitudinal modeling (Figure 1, red metabolites). Specifically, we found two significantly down-regulated metabolites, 3-hydroxyoleylcarnitine (C18_1-OH) (*p* < 0.001) and citrulline/phenylalanine ratio (Cit/Phe) (*p* < 0.001); and two significantly up-regulated metabolites, phenylalanine/tyrosine ratio (Phe/Tyr) (*p* < 0.001) and octanoylcarnitine/decanoylcarnitine ratio (C8/C10) (*p* < 0.001), in the NEC cohort compared to the controls (Figure 2).

Newborns who later developed NEC received significantly less enteral feeding and a predominance of nutrition by parenteral route (IV). Newborns who developed NEC also received significantly lower weight-adjusted total calories over the first six weeks of life (Figure 3, Panel A). These differences remained significant even when data obtained after NEC diagnosis were removed from the analysis (Figure 3, Panel B). Each of the described differences in nutritional parameters are apparent with divergence between future NEC cases and control newborns between day 7 and 10 (Figure 3, Panel B).

Specific time point analyses demonstrated that infants who developed NEC began receiving significantly lower weight-adjusted total calories by post-delivery day 7 and this difference persisted at day 28 (Table 3)—at which time the majority of cases of NEC (56/73) had been diagnosed. Case and control newborns received similar amounts of IV nutrients (AAs, lipids, glucose) before day 28, whereas case newborns with NEC received significantly more IV nutrients after day 28. Infants who developed NEC received significantly lower volumes of enteral feedings by day 7, and that trend persisted through day 42.

Newborns who later developed NEC were started on enteral feeds later (4.2 vs. 2.5 days, *p* ≤ 0.001) and took longer to achieve full enteral feeds (30.6 vs. 18.3 days, *p* ≤ 0.001). The proportion of enteral feedings (mL/kg/day) given as human breast milk (mL/kg/day) was not significantly different at any time point: day 1 (0.58 vs. 0.58, *p* = 0.50), day 7 (0.82 vs. 0.81, *p* = 0.56), day 28 (0.65 vs. 0.60, *p* = 0.74), and day 42 (0.44 vs. 0.52, *p* = 0.16). Despite receiving significantly less overall weight-adjusted calories than control newborns, case newborns had similar weight gain velocities (in g/kg/day) for each time interval: day 1–7: −5.99 vs. −5.10, *p* = 0.34; Day 7–28: 21.1 vs. 20.0, *p* = 0.89; Day 28–42: 18.1 vs. 19.8 *p* = 0.055.

## 4. Discussion

In this study, we reveal that premature newborns who develop NEC demonstrate distinct metabolic differences both immediately at birth and within the first 42 days of life. Newborns who developed NEC received fewer weight-adjusted calories and less enteral feeding prior to onset of clinical disease. These results support the hypothesis that NEC is associated with metabolic derangement as reflected in newborn screen analytes and may be compounded by clinical nutrition practice variability that occurs prior to NEC diagnosis.

The number of specific metabolites that were associated with a risk of NEC increased with each successive time interval analysis indicating progressive metabolic deviation between newborns who develop NEC and those who do not over time. The observed metabolite differences paralleled specific trends in nutritional delivery: the infants who developed NEC received more total parenteral nutrition (TPN) and less enteral feeds over the course of observation. It is essential to emphasize that the described differences in metabolism and nutrition were present prior to any clinical suspicion or diagnosis of disease according to stringent criteria: the requirement for surgical confirmation or radiographic evidence of definitive NEC (i.e., pneumatosis intestinalis, hepatobiliary gas, or pneumoperitoneum). Whether the progressive metabolic differences led to enteral feeding intolerance requiring dependence on TPN or differences in nutritional delivery resulted in changes in nutrient metabolic processing remains unknown yet represents an important focus open to additional study.

The pattern of specific metabolite differences is also revealing. At the early time points in the analyses (day 1, 7), the differences involved predominantly AAs, whereas at the later time points (day 28, 42), the differences were predominantly ACs. It is common clinical practice in the care of preterm infants to start IV AAs on day 1–2 of life and delay administration of IV lipids for several days—practices that may in part explain the predominance of NEC-associated AAs at the early time points. It is important to note, however, that the first newborn sample (day 1) was obtained within 24 h of life for both cases and controls. Both groups of newborns were receiving approximately the same volume of IV nutrients (AAs included) and total weight adjusted calories at the time of the first and second metabolic screen. This suggests that both groups would have been subject to nearly equivalent nutritional exposures and yet significant differences were observed. Thus, these observations support the hypothesis of inherent increased metabolic susceptibility in premature infants at birth as previously reported and built upon with the findings reported here [14].

Several AAs that were associated with NEC in this study merit additional consideration given the interesting potential pathophysiological implications. Citrulline and arginine, both non-essential AAs, have received scrutiny in the NEC literature due to the relationship to nitric oxide (NO) synthesis—a reaction in which arginine is converted to NO and L-citrulline by nitric oxide synthase (NOS). In contrast to our findings of higher levels of arginine on days 1 and 7, several small cohort studies demonstrated reduced serum levels of arginine in infants who developed NEC [18,19]. It has been suggested that reduced availability of arginine in these infants resulted in decreased NO synthesis leading to decreased perfusion and tissue hypoxemia contributing to the intestinal injury observed in NEC. More recent studies have identified an inducible form of NOS that is upregulated in inflammatory states, including NEC [20]. High levels of NO observed during inflammation may detrimentally affect intestinal barrier and mitochondrial function secondary to generation of reactive oxygen species. As such, the theory that insufficient arginine substrate for NO synthesis contributes to the pathogenesis of NEC is plausible but the mechanistic link remains one of speculation (Figure 4) suggesting the need for additional research to reveal the precise role of arginine and citrulline metabolism in NEC pathophysiology.

Phenylalanine is an essential AA that is used predominately as a precursor for tyrosine synthesis, a reaction medicated by phenylalanine hydroxylase using 5,6,7,8-tetrahydrobiopterin (BH4) as a cofactor. BH4 is depleted under conditions of oxidative stress preventing the catalyzation of phenylalanine to tyrosine [21]. The observation of higher levels of phenylalanine in the infants who developed NEC is particularly relevant given that tyrosine levels were not found to be independently associated with NEC in our analyses. Tyrosine is an AA that is administered as part of IV AA infusions and its absence in our analysis suggests that the metabolic defect responsible for the differences observed in the NEC cohort is upstream of tyrosine, i.e., in the conversion of phenylalanine to tyrosine. Interestingly, BH4 is also a cofactor for NO synthesis and with insufficient cofactor supply, NOS will generate reactive oxygen species instead of NO. The findings of elevated arginine levels on days 1 and 7, as well as a reduced citrulline to phenylalanine ratio on day 7 could occur if there was decreased conversion of arginine to NO due to a BH4 deficiency in the immediate neonatal period (Figure 2, Figure 3 and Figure 4). Taken together, these observations may reflect a state of systemic oxidative stress in these infants from birth leading to depletion of BH4 levels, the observed metabolic differences, and the generation of oxygen free radicals that contribute to the intestinal damage seen in NEC (Figure 4) [22].

Reduced postnatal weight gain has been associated with increased risk of NEC, bronchopulmonary dysplasia, late onset sepsis, and unfavorable neurodevelopmental outcomes [23,24]. Lower total calories delivered over the first few weeks of life has also been specifically linked to increased risk of ROP in premature infants [25,26]. We have demonstrated a deficit in total caloric delivery in infants who develop NEC that was not accompanied by a decrease in growth velocity, a finding that has not previously been reported. This suggests that decreased caloric delivery has a more subtle impact on neonatal development and metabolism that is not captured by anthropomorphic measurements, including weight. Particularly in the metabolically vulnerable preterm infant, a deficient energy balance can lead to bioenergetic compromise at the cellular level contributing to organ injury and the development of acquired diseases of prematurity, including NEC [13]. Based on recent literature linking caloric deficiencies to disease of prematurity, a calorie goal of at least 115 kcal/kg/day has been proposed [13]—a target that was not achieved at any time point by the cohort of infants with NEC in our study. Additionally, individual macronutrient delivery (carbohydrate, lipid, and protein) is also relevant to disease risk and onset, however, we lacked sufficiently detailed data to further evaluate these variables with greater precision [25,26]. Clinicians currently lack the tools to precisely evaluate energy balance and nutritional effectiveness relative to risk of acquired disease in preterm infants representing a substantial obstacle in neonatal care that precludes evidence-based nutritional interventions.

A limitation of this study is the lack of more detailed nutrition data. Nutrition data were averaged over each of the time intervals preventing the evaluation of daily caloric intake on the risk on developing NEC and likely resulting in an attenuation of estimated risks. Future studies will be needed to evaluate in more depth the role of daily caloric and macronutrient delivery on disease outcomes and serum metabolites. Further, given the findings that suggest a link between BH4 levels and NEC, additional nutritional variables and metabolites, including specific vitamins and minerals, should be the subject of future studies. Finally, additional studies to determine the cause or consequence for significant differences in enteral feeding and associated metabolic changes that occur with time prior to NEC diagnosis are needed.

## 5. Conclusions

The most premature newborns, and those that are destined to develop NEC during early postnatal life, appear to be metabolically different or impaired beginning at birth. This is reflected in the significantly different metabolic patterns measured in the first metabolic screen prior to the subsequent influence of exogenous nutrition supplementation. Newborns destined to develop NEC appear to have clinical signs of early enteral feeding intolerance as reflected in the significant difference in enteral feedings (total enteral calories—weight adjusted), and consequent need for greater proportion of total calories given parenterally. Given the pattern of accumulating AA and AC differences longitudinally in the NEC cases suggests the need for clinicians to be exquisitely cognizant of feeding intolerance in at risk newborns and the possible metabolic derangements imparted through the increased reliance on TPN to maintain total calories. Currently, clinicians do not have the tools to fully understand the potential detrimental impact on overall clinical outcomes reflected in these metabolic abnormalities described herein. Since there was no difference in growth velocity between the NEC cases and controls, additional study is needed to determine qualifying measures of effective nutritional support beyond anthropometrics that are sensitive to these more subtle biochemical changes relative to the balance of enteral feeding and IV macronutrients and should include measures of potential biologic benefit and harm.

## Figures and Tables

**Figure 1 nutrients-12-01275-f001:**
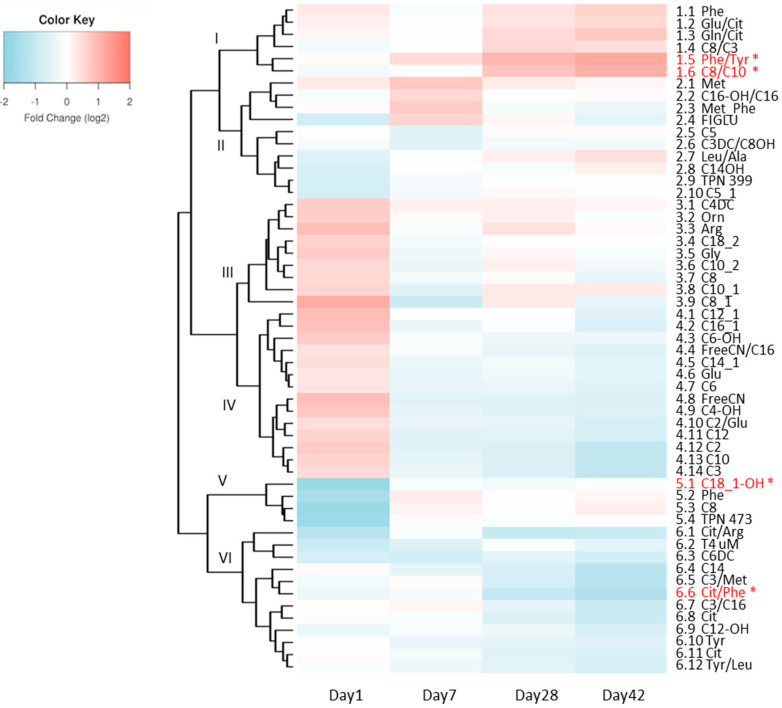
Heat map depicting the fold change in the ratio (log2FC) of all metabolites and respective ratios at each time point. * Metabolites that remained significant in longitudinal modeling after Bonferroni correction are depicted in red.

**Figure 2 nutrients-12-01275-f002:**
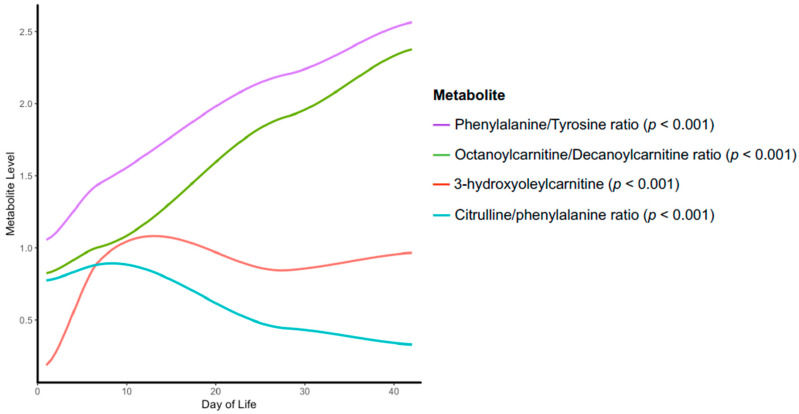
Loess curve depicting time trend of those metabolite relationships that are statistically significant upon longitudinal modeling and Bonferroni correction.

**Figure 3 nutrients-12-01275-f003:**
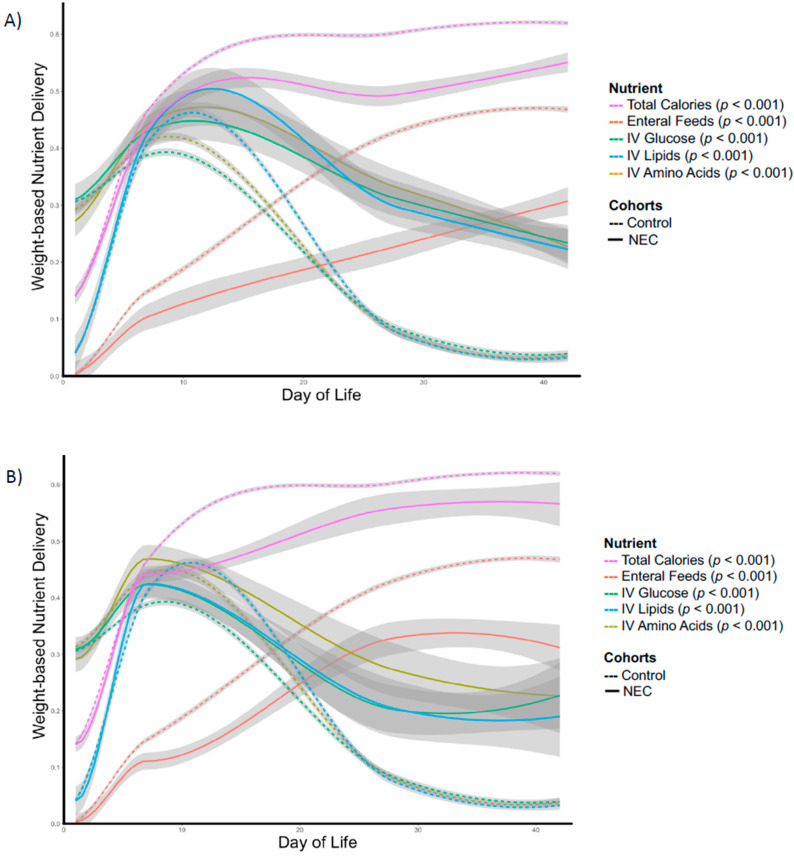
Longitudinal comparison of nutrients administered over the first 6 weeks of life between necrotizing enterocolitis (NEC) case and control infants: (**A**) Entire NEC cohort, (**B**) NEC cohort using only data obtained prior to NEC diagnosis.

**Figure 4 nutrients-12-01275-f004:**
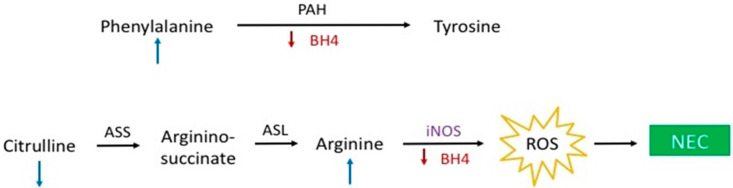
Potential hypothesis outlining the role of tetrahydrobiopterin (BH4) and nitric oxide (NO) in the pathophysiology of necrotizing enterocolitis (NEC).

**Table 1 nutrients-12-01275-t001:** Analytes that are associated with an increased risk of necrotizing enterocolitis (NEC) by day. Adjusted for birth weight and gestational age, transfused (Y/N), postnatal steroids (Y/N), late onset sepsis, and retinopathy of prematurity.

Day Post-Delivery	Metabolite	OR	95% CI	*p*-Value
Day 1	Alanine	1.002	1.0004–1.004	0.01
	Phenylalanine	1.01	1.001–1.01	0.01
	Free Carnitine	1.02	1.002–1.04	0.03
	C16	1.60	1.04–2.44	0.03
	Arginine	1.01	1.001–1.02	0.03
	C14:1/C16	0.87	0.75–0.98	0.04
	Citrulline/Phenylalanine	0.63	0.40–0.96	0.04
Day 7	Methionine/Phenylalanine	1.07	1.01–1.12	0.01
	Methionine	1.01	1.001–1.02	0.02
	Citrulline/Arginine	0.92	0.84–0.98	0.02
	Arginine	1.04	1.003–1.07	0.03
	C12-DC	0.85	0.73–0.98	0.03
	Leucine/Alanine	0.88	0.78–0.99	0.04
Day 28	C5/C4	1.23	1.06–1.43	0.01
	C2	0.90	0.80–0.99	0.04
	C4-OH	0.27	0.07–0.85	0.04
Day 42	C8/C16	1.93	1.14–3.41	0.02
	C4DC	3.42	1.11–10.59	0.03
	C10:1	1.12	1.01–1.25	0.03
	C12-OH	2.48	1.04–6.39	0.4
	C10:2	1.54	1.01–2.40	0.04

**Table 2 nutrients-12-01275-t002:** Demographics and characteristics of cases and controls.

Characteristics	NEC * (*n* = 73)	Control (*n* = 814)	*p*-Value (NEC vs. Control)
Females, *n* (%)	39 (53.4%)	406 (49.9%)	0.56
Race, *n* (%)			0.54
Black	27 (37.0%)	197 (24.2%)	
White	29 (39.7%)	496(60.9%)	
Asian	2 (2.7%)	23 (2.8%)	
Other	15 (20.5%)	98 (12.0%)	
Birth weight (g), median (IQR)	880 (709, 1180)	1090 (860, 1303)	<0.001
Gestational age (weeks), median (IQR)	26 (25, 29)	28 (27, 30)	<0.001
Age at NEC diagnosis (days), median (IQR)	20.5 (12.75, 28)	--	--
Apgar at 1 minute (minutes), mean (IGR)	5.2 (4, 7)	5.2 (3, 7)	0.91
Apgar at 5 minutes (minutes), mean (IQR)	7.2 (6, 9)	7.3 (7, 9)	0.47
Antenatal Steroids, *n* (%)	65 (89.0%)	758 (88.8%)	0.94
Transfused, *n* (%)	66 (90.4%)	462 (54.1%)	<0.001
Postnatal steroids, *n* (%)	20 (27.4%)	92 (10.8%)	<0.001
Late onset sepsis, *n* (%)	23 (31.5%)	88 (10.3%)	<0.001
Retinopathy of prematurity, *n* (%)	26 (36.6%)	211 (24.9%)	0.030
Intraventricular hemorrhage, *n* (%)	16 (21.9%)	184 (21.5%)	0.94
Bronchopulmonary dysplasia, *n* (%)	32 (43.8%)	292 (34.2%)	0.097
Patent ductus ateriosus, *n* (%)	35 (48.6%)	343 (40.5%)	0.18

* NEC—necrotizing enterocolitis.

**Table 3 nutrients-12-01275-t003:** Individual time point analysis of nutritional variables using only data obtained before necrotizing enterocolitis (NEC) diagnosis.

	Day 1			Day 7			Day 28			Day 42		
	NEC	Control	*p-*Value	NEC	Control	*p-*Value	NEC	Control	*p-*Value	NEC	Control	*p-*Value
Total Calories (kcal/kg/d)	33.8	33.9	0.48	87.5	91.6	0.056	108.3	116.2	0.026	109.6	119.1	0.059
IV Amino Acids (g/kg/d)	1.7	1.8	0.077	2.7	2.6	0.14	1.6	0.5	<0.001	1.3	0.2	<0.001
IV Lipids (g/kg/d)	0.2	0.2	0.47	2.0	1.9	0.30	1.0	0.3	0.002	0.9	0.2	<0.001
IV Glucose (g/kg/d)	7.2	7.1	0.35	9.9	9.1	0.069	4.7	1.9	0.003	5.3	0.0	<0.001
Enteral Feeds (mL/kg/d)	1.1	1.1	0.42	33.7	45.0	0.025	100.0	131.0	0.003	94.8	142.2	<0.001

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
