# Peer review of "Progressive Metabolic Dysfunction and Nutritional Variability Precedes Necrotizing Enterocolitis"

_nutrients, 2020, doi:10.3390/nu12051275_

Round 1

Reviewer 1 Report

This is an original and   good paper, suggesting a novel pathophysiologic basis for NEC.

However, I have the following major concerns:

1) The authors say  (line 151) "Newborns who later developed NEC received significantly less enteral feeding and a predominance of nutrition by parenteral route"

How can they explain this? Perhaps the preterm newborns who developed NEC  showed an earlier  enteral feeding intolerance? It could be very useful to evaluate gastric residuals to demontrate this possible relationship.

2) Which type of enteral feeding did the preterm newborns receive during the period of study? Fresh Human Milk, Donor Milk, Special preterm formula?I think it's important to specify this aspect because it's possible that different type of feeding can determine important differences in aminoacids metabolism

3) Can the authors better underline the practical consequences of their discoveries? To change parenteral nutrition composition for preterm newborns during the first days of life? 

Reviewer 2 Report

This is a retrospective multicenter (23 centres in 17 states) study with data from a database with data from a cohort of premature baby's (2009-2012) in whom dried blood spots were available from D1, D8, D 28 and D42. Clinical data were retrieved from the database and results of AA and AC analysis on dry blood spots in cases (diagnosis of NEC, medical and surgical) were compared to controls, in order to find differences preceding the diagnosis of NEC.

It is not clear whether the analysis on the dry blood spots was performed at the time of collection or more recently. In case of the latter, it should be specified how these dry spots were conserved. In case of the former, it is strange that the analysis follows 8 years after the collection of data.

My most important methodological problem with the study, is the fact that no detailed information is available on nutrition in this cohort: AA and AC levels are known to be influenced by quantity and quality of proteins and lipids (also carbohydrates) administered. When looking for associations of metabolic disturbances with NEC, the authors should have controlled for this confounding variable. There is a clear difference in the need for parenteral nutrition between both groups, which could possibly explain the difference in metabolites, except on D1. Although, as expected, more cases of NEC are found in the more premature group, the analysis is neither adjusted for age as a confounding variable. Hence , it seems difficult to conclude whether metabolic differences described are cause, consequence or coïncidence.

In the clinical data, no information on occurence of asfyxia is given (also a known risk factor for NEC).

As this is a multicenter study, more information should be included regarding nutritional protocols in the different centres (were these similar? How about own mothers' milk versus donor milk? which kind of formula-feeding? ) and regarding the diagnostic and therapeutic protocols of NEC. Were NEC cases uniformly distributed across the centres (taking into account confounding variables such as gestational age, …)?

The paper lacks a clear conclusion as to what the clinical relevance of the findings (which should be corrected for gestational age and nutritional intake) is, and what kind of (clinical) research would be necessary to test the hypotheses raised by this descriptive work. Discussion and conclusion need adaptation including these important issues.

Round 2

Reviewer 1 Report

The authors improved the paper and answered to some questions. However, they must better clarify the feeding protocol in each center. It's the only concern I still have

Reviewer 2 Report

The authors have adequately adressed the questions and clarifications asked for in the first report and have adapted their conclusion.

The only major methodological point that still requires specification is the feeding protocol in the different centers. The authors have now added the following information (line 79-80): "Standardized fluid management and nutritional guidelines were utilised at each site", without a reference. To me, it remains unclear whether all centres did use the same nutritional protocol and the same formula milk and/or breast milk fortifier? As this information is important in order to interpret the results and consider their clinical importance, the authors should either refer to the protocol used uniformly in all centers, or describe it in more detail.

The information regarding the PN and AA solutions included in the response letter, is preferably also mentioned succintly in the manuscript.
